# Reduced Nitric Oxide Synthase Involvement in Aigamo Duck Basilar Arterial Relaxation

**DOI:** 10.3390/ani13172740

**Published:** 2023-08-28

**Authors:** Siyuan Wu, Tomoki Ootawa, Ryoya Sekio, Henry Smith, Md. Zahorul Islam, Ha Thi Thanh Nguyen, Yasuhiro Uno, Mitsuya Shiraishi, Atsushi Miyamoto

**Affiliations:** 1Department of Basic Veterinary Science, Joint Graduate School of Veterinary Medicine, Kagoshima University, 1-21-24 Korimoto, Kagoshima 890-0065, Japan; 2Japan Wildlife Research Center, 3-3-7 Kotobashi, Tokyo 130-8606, Japan; 3Department of Veterinary Pharmacology, Joint Faculty of Veterinary Medicine, Kagoshima University, 1-21-24 Korimoto, Kagoshima 890-0065, Japan; 4Department of Pharmacology, Faculty of Veterinary Science, Bangladesh Agricultural University, Mymensingh 2202, Bangladesh; 5Department of Veterinary Pharmacology and Toxicology, Faculty of Veterinary Medicine, Vietnam National University of Agriculture, Gia Lam, Hanoi 131000, Vietnam

**Keywords:** cerebral artery, duck, endothelium, highly pathogenic avian influenza, nitric oxide

## Abstract

**Simple Summary:**

The basilar artery is a vital cerebral blood vessel common in most vertebrates and constantly supplies blood to the hindbrain where many vital functions are coordinated. Avian basilar arterial responsiveness to vasoactive substances has been characterized only in chickens. In this artery, the endothelium plays an important role in relaxation, and endothelial dependence may explain the lethality of the highly pathogenic avian influenza virus, which reportedly induces apoptosis in the cerebrovascular endothelium. Our present results in ducks suggest a contrast to the previously reported results in chickens with regard to basilar arterial relaxation: The involvement of endothelial nitric oxide as a relaxing factor appears to be reduced in duck basilar arteries. Our research may help scientists to better understand the resistance to the highly pathogenic avian influenza virus that may be conferred by the cerebrovascular endothelium in ducks.

**Abstract:**

The basilar arterial endothelium mediates blood vessel relaxation partly through the release of nitric oxide (NO). Apoptosis of cerebrovascular endothelial cells is linked to a high mortality rate in chickens infected with the highly pathogenic avian influenza virus, but interestingly, ducks exhibit a greater resistance to this virus. In this study, we examined the responsiveness of duck basilar arteries (BAs) to various vasoactive substances, including 5-hydroxytryptamine (5-HT), histamine (His), angiotensin (Ang) II, noradrenaline (NA), acetylcholine (ACh), and avian bradykinin ornithokinin (OK), aiming to characterize the receptor subtypes involved and the role of endothelial NO in vitro. Our findings suggest that arterial contraction is mediated with 5-HT_1_ and H_1_ receptors, while relaxation is induced with β_3_-adrenergic and M_3_ receptors. Additionally, OK elicited a biphasic response in duck BAs, and Ang II had no effect. Endothelial NO appears to be crucial in relaxation mediated with M_3_ and OK receptors but not β_3_-adrenergic receptors in the duck BA. The reduced endothelial NO involvement in the receptor-mediated relaxation response in duck BAs represents a clear difference from the corresponding response reported in chicken BAs. This physiological difference may explain the differences in lethality between ducks and chickens when vascular endothelial cells are infected with the virus.

## 1. Introduction

Ducks appear to be markedly more resilient to highly pathogenic avian influenza virus (HPAIV) than chickens [1,2]. For example, in one study, all wild ducks (from the genus *Anas*) experimentally infected with HPAIV remained free of symptoms and survived [3]. In chickens, by contrast, HPAIV infection often leads to fatal outcomes in which the vascular endothelium may be implicated. Vascular endothelial cells in the liver, kidney, and brain in chickens with HPAIV are reportedly prone to apoptosis, and this phenomenon has been linked to the deaths of infected birds [4,5]. Acute apoptosis is reportedly associated with high levels of endothelial nitric oxide (NO) release during HPAIV infection, and the released NO may act with other damaging oxidants to promote excessive inflammation [6]. Reports from human physiology suggest that elevated NO levels can promote proinflammatory effects, including increased vascular permeability, cytotoxicity, and inflammatory cell infiltration [6,7,8], and any related loss of endothelial function in vital arteries could quickly prove fatal. Thus, it is plausible that ducks’ resilience to HPAIV may stem from distinctive characteristics of their vascular endothelium, but vasoreactivity and endothelial NO release remain largely uncharacterized in duck species.

The basilar artery (BA) may provide a useful location for investigating vasoreactivity and endothelial release in ducks. It is a vital artery that constantly supplies blood to the hindbrain and a common cerebrovascular feature across vertebrates that contributes to the maintenance of cerebral circulatory volume [9]. Its role in birds appears to be the same as that in mammals, although chickens are the only avian species in which BA reactivity has been studied [10,11,12]. In chickens, the BA endothelium has been identified as a powerful source of spontaneous NO release, which was based on strong contractions noted after the application of N^ω^-nitro-L-arginine (L-NNA), a NO synthase (NOS) inhibitor [11]. Furthermore, previous investigations of endothelial cell receptors have indicated that beta-3 adrenergic receptors (β_3_-ARs), histamine (His) H_1_ receptors, and muscarinic acetylcholine (ACh) M_3_ receptors are involved in the relaxation of the chicken BA via the NO pathway, suggesting this arterial relaxation is strongly dependent on the endothelial release of NO [10,11,12]. 

Evolutionarily speaking, ducks and chickens are separated by 90 million years, and avian genomes tend to be more strongly conserved than mammalian genomes [13,14]. We thus considered it plausible that ducks may have a less complex vascular endothelium than chickens, who were later to emerge as a species. Interestingly, HPAIV reportedly rarely shows endothelial tropism in either wild or domestic ducks [15,16].

Against this background, we aimed to investigate the vascular endothelium of the BA in ducks by characterizing its capacity for endothelial NO release and its response to a range of vasoactive substances previously evaluated in our studies of this artery in chickens.

## 2. Materials and Methods

### 2.1. Tissue Preparation

BAs were obtained from freshly slaughtered Aigamo ducks (*Anas platyrhynchos*/*Anas platyrhynchos var. domesticus* hybrid; *n* = 71, both sexes, weight: 2.97 ± 0.04 kg). The ducks had been raised for meat and allowed to forage on insects in rice paddies on farms in Kagoshima Prefecture, Japan, and were slaughtered in accordance with the relevant Japanese law on agricultural animals. The sampled arteries were transferred to our laboratory in ice-cold physiological saline (119 mM NaCl, 4.7 mM KCl, 1.6 mM CaCl_2_, 1.2 mM MgCl_2_, 25 mM NaHCO_3_, 1.2 mM KH_2_PO_4_, and 10 mM glucose, pH 7.4) aerated with carbogen [95% (vol/vol) O_2_, 5% (vol/vol) CO_2_]. Each artery was immediately dissected free of adherent tissues under a stereomicroscope. All experiments were performed in accordance with the Guidelines for Animal Experiments of Kagoshima University. Our data were obtained from Aigamo ducks, not from species-pure ducks, and this represents a limitation of this study.

### 2.2. Reagents

The following reagents were used at the described concentrations: His hydrochloride (10^−6^–10^−3^ M, Sigma-Aldrich, St. Louis, MO, USA), isoproterenol (10^−9^–10^−5^ M, Sigma-Aldrich), ketanserin tartrate (10^−6^ M, Sigma-Aldrich), methiothepin maleate (10^−8^–10^−6^ M, Sigma-Aldrich), diphenhydramine hydrochloride (10^−7^–10^−5^ M, Sigma-Aldrich), cimetidine (10^−5^ M, Sigma-Aldrich), angiotensin (Ang) II (human) acetate salt (10^−8^–10^−4^ M, Sigma-Aldrich), butoxamine hydrochloride (10^−6^ M, Sigma-Aldrich), SR 59230A (10^−7^–10^−6^ M, Sigma-Aldrich), L-NNA (10^−4^ M, Sigma-Aldrich), noradrenaline (NA, 10^−9^–10^−4^ M, Tokyo Chemical Industry, Tokyo, Japan), phentolamine mesylate (10^−5^ M, Tokyo Chemical Industry), 5-hydroxytryptamine (5-HT)-creatinine sulfate (10^−8^–10^−4^ M; Merck, Darmstadt, Germany), adrenaline (10^−9^–10^−5^ M, Daiichi Sankyo, Tokyo, Japan), ACh chloride (10^−9^–10^−5^ M, Daiichi Sankyo), procaterol (10^−9^–10^−5^ M, Fujifilm-Wako, Tokyo, Japan), atenolol (10^−6^–10^−5^ M, LKT Laboratories, Tokyo, Japan), hexahydro-sila-difenidol hydrochloride, *p*-fluoroanalog (pFHHSiD, 10^−7^, 10^−6^ M; Research Biochemical, Natick, MA, USA), avian bradykinin ornithokinin (OK, 10^−9^–10^−5^ M, BEX Co. Ltd., Tokyo, Japan), and sodium nitroprusside (SNP, 10^−4^ M; Nacalai Tesque, Kyoto, Japan). All reagents were used in accordance with the relevant manufacturer’s instructions. Each of the selected agonists had previously been validated in vertebrate species (including avian species) [17,18,19,20]. Their reported effects on cerebral arteries in a range of species as demonstrated in vitro studies are shown in Table 1.

### 2.3. Functional Study

The methods applied in the functional study were as described in our previous study in chickens [12]. Briefly, three or four rings, approximately 2 mm wide, were cut from each duck BA. Each ring was mounted horizontally between two L-shaped, stainless-steel holders (outer diameter 0.03 mm) with one part fixed to an isometric force transducer. The mounted ring was then immersed in a 4 mL water-jacketed micro-tissue organ bath (UMTB-1, Unique Medical Co. Ltd., Tokyo, Japan) containing oxygenated physiological saline at 41 °C (pH 7.4). For experiments with precontraction, 5-HT was selected as the inducing agent. Isometric tension was measured with an amplifier. 

The log concentration ratio of EC_50_ values (i.e., the concentration producing a half-maximal response) in the absence or presence of an antagonist was calculated and plotted against the logarithm of the antagonist concentration to obtain the pA_2_ values [30].

At the end of the relaxant response, SNP (10^−4^ M) was applied to produce maximal relaxation, which was taken as 100%. 

### 2.4. Statistical Analysis

Results are expressed as the mean ± SEM, and statistical analyses were performed using Student’s *t*-test or the Bonferroni test after one-way ANOVA (Stat View J-4.5, Abacus Concepts Inc., Berkeley, CA, USA). *p*-values < 0.05 were considered statistically significant.

## 3. Results

### 3.1. Spontaneous Nitric Oxide and Prostaglandin Release

The typical responses to L-NNA (a NOS inhibitor, 10^−4^ M) with a subsequent application of indomethacin (a cyclooxygenase inhibitor, 10^−5^ M) under the resting tone are illustrated in Figure 1: L-NNA-induced contraction (30.5 ± 5.4% to 60 mM KCl, *n* = 4 ducks) under resting tension and indomethacin-induced relaxation (−36.1 ± 2.1% to 10^−4^ M SNP, *n* = 4 ducks) under contraction induced with L-NNA. The contraction induced with 60 mM KCl was 1.84 ± 0.04 mN.

### 3.2. Responsiveness to Vasoactive Substances

To ascertain the dominant receptor subtypes in duck BAs, we investigated the effect of a range of substances with a known vasoactive effect in chickens and other species. Specifically, the agents used to generate effects on the vascular endothelium were 5-HT, His, Ang II, ACh, NA [a beta-1 (β_1_) and non-selective alpha (α) AR agonist], and OK (an avian bradykinin receptor agonist). The contractile response was measured under a resting tension condition. Under this condition, 5-HT and His induced contraction in concentration-dependent manners with respective half-maximal effective concentrations (logarithmically adjusted, hereafter, pEC_50_ values) of 5.84 ± 0.06 and 4.37 ± 0.09, whereas no effect was noted for NA in the absence or presence of propranolol (a non-selective β-AR antagonist, 10^−5^ M) or Ang II (Figure 2A and Table 2). The relaxation response was measured under precontraction with 5-HT. Following precontraction, ACh, NA, and OK induced relaxation in a concentration-dependent manner, yielding respective pEC_50_ values of 6.18 ± 0.17, 6.28 ± 0.05, and 6.20 ± 0.25. Phentolamine (10^−5^ M), a non-selective α-AR antagonist, had no significant effect on the response to NA (Figure 2B and Table 2).

### 3.3. Involvement of 5-Hydroxytryptamine Receptor Subtype

To ascertain the predominant 5-HT receptor subtype in the duck BA, we investigated the effects of methiothepin (a non-selective 5-HT receptor antagonist) and ketanserin (a 5-HT_2_ receptor selective antagonist) on 5-HT-induced contraction. The Schild plot for methiothepin showed a slope of 1.16 ± 0.19 (Figure 3C), which did not significantly diverge from unity. Its calculated pA_2_ value was 8.52 ± 0.17, whereas ketanserin (10^−6^ M) had no significant effect on the 5-HT-induced contraction in the duck BAs (Figure 3B).

### 3.4. Involvement of Histamine Receptor Subtypes

To ascertain the predominate His receptor subtype in the duck BA, we investigated the effects of diphenhydramine (an H_1_ receptor antagonist) and cimetidine (an H_2_ receptor selective antagonist) on His-induced contraction. The Schild plot for diphenhydramine (10^−7^–10^−5^ M) showed a slope of 1.09 ± 0.15 (Figure 4B), which did not significantly diverge from unity. Its calculated pA_2_ value was 6.89 ± 0.13, whereas cimetidine (10^−5^ M) had no significant effect (Figure 4).

### 3.5. Responsiveness to β-Adrenergic Receptor Agonists

To elucidate the roles of β-ARs in the duck BA, we applied different β-AR agonists that may elicit relaxation effects and monitored the vascular response. The agents applied were isoproterenol (a non-selective β-AR agonist), NA (a β_1_-AR and non-selective α-AR agonist), adrenaline (a non-selective β-AR and non-selective α-AR agonist), and procaterol (a β_2_-AR agonist). Three of the four β-AR agonists induced concentration-dependent relaxation of the duck BAs; the exception was procaterol (Figure 5). Their rank order of potency as inducers of relaxation was isoproterenol > NA > adrenaline (Table 2). None of these agents induced contraction.

### 3.6. Involvement of β-Adrenergic Receptor Subtype

To ascertain the predominant β-AR subtype in the duck BA, we investigated the effects of atenolol (a β_1_-AR antagonist), butoxamine (a β_2_-AR antagonist), and SR 59230A (a β_3_-AR antagonist) on isoproterenol-induced relaxation. In this experiment, atenolol (a β_1_-AR antagonist, 10^−6^ M) and butoxamine (a β_2_-AR antagonist, 10^−6^ M) had no significant effect on isoproterenol-induced relaxation (Figure 6A). In contrast, only SR 59230A effectively antagonized the isoproterenol-induced relaxation, and its slope of the Schild plot was 1.18 ± 0.11, which did not significantly diverge from unity (Figure 6C). Its calculated pA_2_ value was 7.03 ± 0.08. Furthermore, L-NNA (10^−4^ M) did not affect the isoproterenol-induced relaxation in duck BAs (Figure 6A).

### 3.7. Involvement of Muscarinic Receptor Subtype

To ascertain the predominant muscarinic ACh receptor subtype in the duck BA, we investigated the effects of atropine (a non-selective muscarinic ACh receptor antagonist), pirenzepine (a muscarinic ACh M_1_ receptor selective antagonist), methoctramine (a muscarinic ACh M_2_ receptor selective antagonist), and pFHHSiD (a muscarinic ACh M_3_ receptor selective antagonist) on ACh-induced contraction. Atropine shifted the concentration-response curve for ACh to the right at 10^−8^ M and largely abolished ACh-induced relaxation at 10^−7^ M (Figure 7A). For selective muscarinic antagonists, the respective slopes of Schild plots for pirenzepine and pFHHSiD were 1.06 ± 0.10 and 0.97 ± 0.08, neither of which significantly diverged from unity. The respective pA_2_ values yielded with pirenzepine and pFHHSiD were 6.52 ± 0.12 and 8.06 ± 0.13 (Figure 7E,F), whereas methoctramine did not affect the ACh-induced relaxation in duck BAs. L-NNA mostly abolished the ACh-induced relaxation of duck BAs (Figure 7D).

### 3.8. Effects of Nitric Oxides Synthase and Cyclooxygenase Inhibitors on Ornithokinin-Induced Response

To investigate OK-induced relaxation in the duck BA, we applied L-NNA and L-NNA plus indomethacin. In the presence of L-NNA (10^−4^ M), OK-induced relaxation was attenuated significantly (*p* < 0.01) and even transferred to contraction at 10^−5^ M, whereas indomethacin (10^−5^ M) abolished this contraction (Figure 8).

## 4. Discussion

The present study is the first to characterize basilar arterial responsiveness to a range of vasoactive substances in ducks. We showed that NO is a diminished endothelial mediator in ducks relative to its reported role in chickens. This information will prove especially useful for comparisons between ducks, who are not prone to disease and death after HPAIV infection, and chickens, who are.

We demonstrated the involvement of spontaneously released endothelial NO based on the contraction induced with the NOS inhibitor L-NNA followed by the relaxation induced with indomethacin, which inhibits cyclooxygenase, under resting tension (Figure 1). Our findings in ducks are similar to the results obtained in some mammalian species, including dogs and bats [26,31]. In porcine BAs, further studies demonstrated that spontaneous NO is released from endothelial cells and spontaneous thromboxane A_2_, which is formed from PGH_2_ and inhibited with indomethacin, is also released from endothelial cells [28,32]. The mechanism for maintaining cerebrovascular tone in ducks partially resembles that in chickens. However, a key difference was that the L-NNA-induced contraction in ducks (30.5%) was markedly weaker than that in chickens (122.1%) [11], suggesting that ducks experience less involvement of spontaneous endothelial NO than chickens.

We also investigated the receptor subtypes involved in contraction and relaxation and their location (smooth muscle or endothelial cells) in the duck BA. 

It appears that 5-HT receptors play a similar role in ducks and chickens, as the receptor agonist induced a concentration-dependent contraction in isolated duck BAs in both the avian species we evaluated, and we found that the 5-HT_1_ receptor may be the dominant receptor of this subtype (Figure 3). Based on an evaluation of the half-maximal effect concentrations, we found that the 5-HT receptor antagonist methiothepin produced a similar effect on the duck BA to that observed on the chicken BA and the rabbit saphenous vein where the vascular response (vasocontraction) is mediated via activation of the 5-HT_1_ receptor [10,33]. Furthermore, the effects of methiothepin and the 5-HT_2_ antagonist ketanserin in the duck BA (respective rightward shifts in the concentration–response curve of 35.5-fold and 2.7-fold) were similar to the corresponding shifts (30-fold and 3-fold, respectively) we previously observed in the chicken BA [10], suggesting ducks and chickens share a similar characterization for this receptor type.

We also profiled the involvement of His receptors. The His-induced, concentration-dependent contraction in isolated duck BAs in this study resembled those observed in porcine, bovine, and equine BAs, with a similar half-maximal effect to that in the bovine report but a smaller half-maximal effect than those in the porcine and equine [24,25] reports. The H_1_-receptor antagonist diphenhydramine (10^−7^–10^−5^ M) had an effect on ducks, but this effect was of a smaller magnitude than those reported in cattle, pigs, and horses [24,25]. The H_2_-receptor antagonist cimetidine had no effect, and we thus consider that H_1_ receptor activation induces contraction of the duck BA. Our findings related to His receptors in ducks present a contrast to the corresponding findings previously reported in chickens. Okuno et al. (2008) [11] reported that both H_1_ and H_2_ receptors, which are located on respective endothelial cells and smooth muscle cells, are involved in the relaxation of the chicken BA. However, in ducks, H_1_ receptors play a dominant role in His-induced contraction, while H_2_ receptors were not involved. 

For adrenergic receptors, noradrenaline had no effect on the BA at resting tension, and phentolamine (a non-selective α-ARs antagonist) had no effect on noradrenaline-induced relaxation under precontraction (Figure 2), suggesting that α-ARs are not involved in duck BA reactivity. This presents a contrast with the chicken BA where α-ARs are present and involved in contraction, and phentolamine inhibits this contraction [12]. For β-ARs, the duck BA appears to possess β_1_-ARs but with a smaller population for this receptor subtype than in chickens based on the similar potency ranking but smaller effect size for isoproterenol and NA [12]. β_2_-ARs appear not to play a major role in the duck BA, although they are present as a non-dominant subtype in chickens based on findings for the effect of procaterol [12]. Interestingly, the effect of the β-AR agonist isoproterenol-induced relaxation was not inhibited with the NOS inhibitor L-NNA (Figure 6A); therefore, we suggest that NO plays no role in β-AR mediation in ducks, which represents a physiological difference from chickens. 

We consider that β_3_-AR is the predominant β-AR subtype in the duck BA based on the results obtained with a range of β-AR antagonists (Figure 6). Most pertinently, only the β_3_-AR antagonist SR 59230A demonstrated an effective antagonistic effect (pA_2_ value: 7.03) in duck BAs comparable to that in rat urinary bladder tissue (7.27) and the guinea pig gastric fundus (7.35) where β_3_-ARs are known to predominate [34,35]. Accordingly, we suggest that β_3_-AR is predominantly involved in isoproterenol-induced relaxation in the duck BA, a phenomenon similar to that reported in chickens. However, this relaxation is endothelium-dependent in chickens but not in ducks. 

Our findings for muscarinic ACh receptors (Figure 7) resembled those observed in bat, chicken, and mouse BAs [10,26,29], suggesting their activation is involved in the response to ACh. We further suggest that the M_3_ receptor is the predominant subtype involved in ACh-induced relaxation, and M_1_ and M_2_ receptors may not be involved in duck BAs. We found no effect for the M_2_ antagonist methoctramine and only a weak effect for the M_1_ antagonist pirenzepine. The latter effect was smaller than that reported in cat cerebral arteries (8.08) where muscarinic M_1_ receptors predominate [36]. However, the M_3_ antagonist pFHHSiD yielded a calculated pA_2_ value (8.06) similar to those reported in bovine coronary arteries (7.64), human uterine arteries (8.17), and chicken BAs (7.55) where the muscarinic M_3_ receptor predominates [10,37,38]. In contrast to our findings on β-ARs, we consider that NO is involved in muscarinic-M_3_-receptor-mediated relaxation in ducks, as it is in other species, including mice, bats, and chickens [10,26,29], based on the abolition of the ACh-induced relaxation with the NO inhibitor L-NNA seen in this study (Figure 7D).

OK, which has been purified from duck plasma, is reported to be an avian bradykinin [39]. We revealed that OK induced relaxation and contraction in duck BAs via regulation of the respective NO and cyclooxygenase pathways. In porcine BAs, bradykinin produced NO and prostaglandin F_2α_ then induced relaxation followed by contraction [32,40]. OK receptors are the avian homolog to mammalian bradykinin receptors and are known to be involved in blood vessel dilatation, smooth muscle contraction, increased vascular permeability, and inflammation [41]. OK receptors reportedly potentiate proinflammatory responses in chicken macrophages [42]. Pharmacological research on these receptors is hampered with a lack of established antagonists; accordingly, it was not possible to investigate the relevant receptor subtypes in this study. However, we anticipate this line of research will be enabled with advances in cell cultures in the future.

BA responsiveness to vasoactive substances is known to differ widely between mammalian species according to differences in receptor subtypes and their distribution on smooth muscle or endothelial cells. To our knowledge, BA responses have never shown fully identical characteristics in any two mammalian species, and we speculate that similar diversity will be observable across avian species based on the present study, which is the second characterization of BA response in an avian species. However, one feature common to ducks in this study and to chickens in our previous report was β_3_-AR-mediated vasodilation, which has never been reported in mammalian cerebrovascular systems [12,21]. We speculate that β_3_-ARs may play an important role in the cardiovascular system of birds, and this represents an interesting line of future research. These physiological differences could be explained with the earlier evolutionary emergence of ducks than chickens, which means that ducks may exhibit simpler endothelial physiology. In contrast, chickens have undergone significant adaptations to suit terrestrial life, leading to more complex functionalities mediated by the endothelium.

## 5. Conclusions

In conclusion, we suggest that the BA of the duck is characterized by a smaller endothelial release of NO and a smaller degree of endothelial involvement in its reactivity than the BA of the chicken. The 5-HT_1_ and H_1_ receptors may be involved in arterial contraction, and the β_3_ and M_3_ receptors may be involved in relaxation. NO plays a role with M_3_ but not β_3_-adrenergic receptors. These physiological differences may help to explain why severe effects (including death) of HPAIV infection are seen in chickens but not ducks.

## Figures and Tables

**Figure 1 animals-13-02740-f001:**
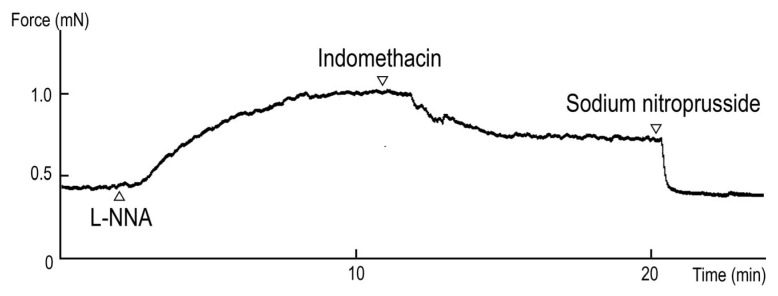
Typical contraction induced with N^ω^-nitro-L-arginine (L-NNA, 10^−4^ M) and relaxation induced with indomethacin (10^−5^ M) under precontracted conditions induced with L-NNA.

**Figure 2 animals-13-02740-f002:**
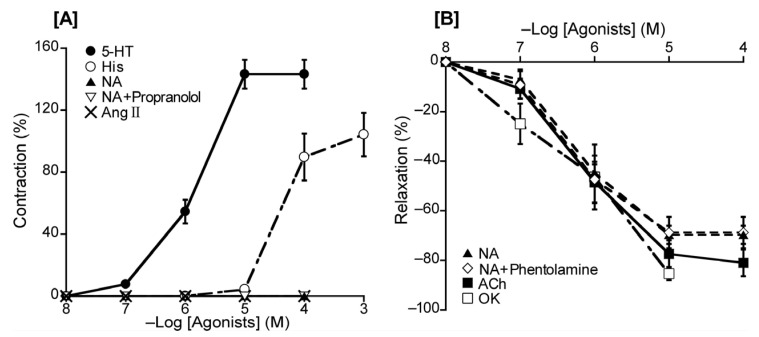
Responsiveness to 5-hydroxytriptamine (5-HT), histamine (His), noradrenaline (NA), and angiotensin (Ang) II under resting tension (**A**) and to NA, acetylcholine (ACh), and ornithokinin (OK) under a precontracted condition induced with 5-HT (**B**). The contraction induced with 60 mM KCl (**A**) and the relaxation induced with 10^−4^ M sodium nitroprusside (**B**) was taken as 100% contraction and relaxation, respectively. Each point represents the mean ± SEM of 4–6 ducks. The percentage of reacting vessels is 100%.

**Figure 3 animals-13-02740-f003:**
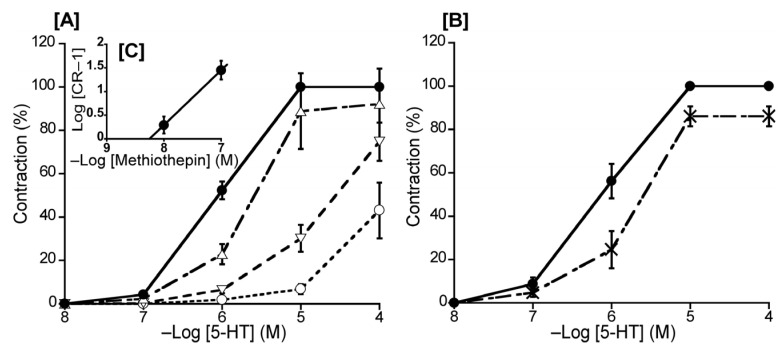
Effects of 5-hydroxytriptamine (5-HT) antagonists on 5-HT-induced contraction in the duck basilar artery. The antagonists were methiothepin (Δ: 10^−8^ M, ▽: 10^−7^ M, ○: 10^−6^ M) (**A**) and ketanserin (×: 10^−6^ M) (**B**). The Schild plot for methiothepin is shown in (**C**). The maximal contraction induced with 5-HT was taken as 100%. Each point represents the mean ± SEM of 6 ducks.

**Figure 4 animals-13-02740-f004:**
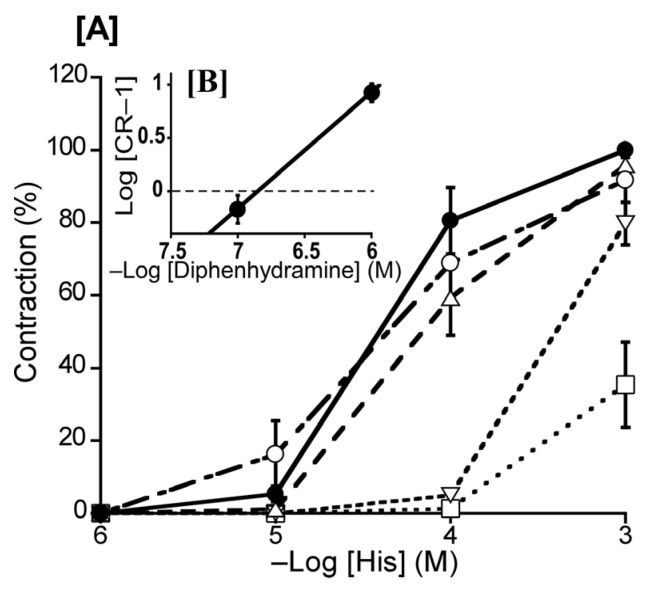
Effects of histamine (His) antagonists on His-induced contraction in the duck basilar artery. The antagonists were diphenhydramine (Δ: 10^−7^ M, ▽: 10^−6^ M, □: 10^−5^ M) and cimetidine (○: 10^−5^ M) (**A**). The Schild plot for diphenhydramine is shown in (**B**). The maximal contraction induced with His was taken as 100%. Each point represents the mean ± SEM of 5 ducks.

**Figure 5 animals-13-02740-f005:**
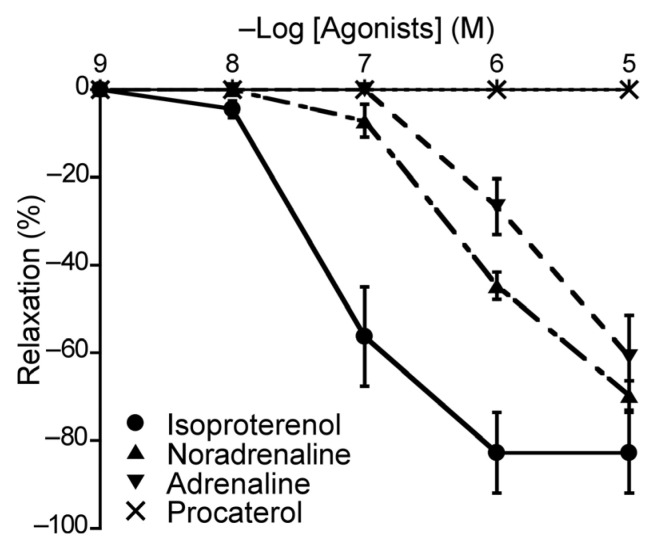
Responsiveness of duck basilar arteries to isoproterenol (●), noradrenaline (▲), adrenaline (▼), and procaterol (×) under precontracted conditions. The relaxation induced with 10^−4^ M sodium nitroprusside was taken as 100%. Each point represents the mean ± SEM of 4–6 ducks.

**Figure 6 animals-13-02740-f006:**
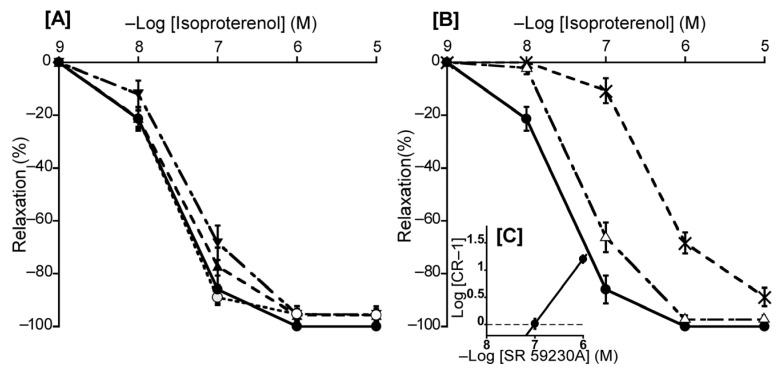
Effects of β-adrenergic receptor antagonists and a NOS inhibitor (L-NNA) on isoproterenol-induced relaxation in the duck basilar artery. The antagonists or inhibitor were atenolol (▲: 10^−6^ M), butoxamine (▼: 10^−6^ M), L-NNA (○: 10^−4^ M) (**A**), and SR 59230A (Δ: 10^−7^ M, ×: 10^−6^ M) (**B**). The Schild plot for SR 59230A is shown in (**C**). The maximal relaxation induced with isoproterenol was taken as 100%. Each point represents the mean ± SEM of 4 or 5 ducks.

**Figure 7 animals-13-02740-f007:**
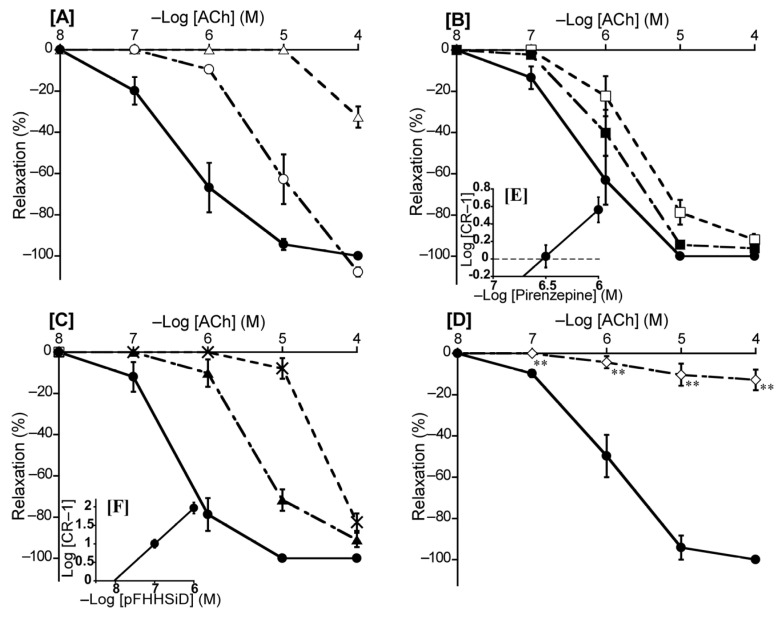
Effects of acetylcholine (ACh) antagonists and a NOS inhibitor (L-NNA) on ACh-induced relaxation in the duck basilar artery. The antagonists or inhibitor were atropine (○: 10^−8^ M, Δ: 10^−7^ M) (**A**), pirenzepine (■: 10^−6.5^ M, □: 10^−6^ M) (**B**), hexahydro-sila-difenidol hydrochloride, *p*-fluoroanalog (pFHHSiD; ▲: 10^−7^ M, ×: 10^−6^ M) (**C**), and L-NNA (◇:10^−4^ M) (**D**). The Schild plots for pirenzepine and pFHHSiD were shown in (**E**,**F**), respectively. The maximal relaxation induced with ACh was taken as 100%. Each point represents the mean ± SEM of 5 ducks. (** *p* < 0.01 vs. control).

**Figure 8 animals-13-02740-f008:**
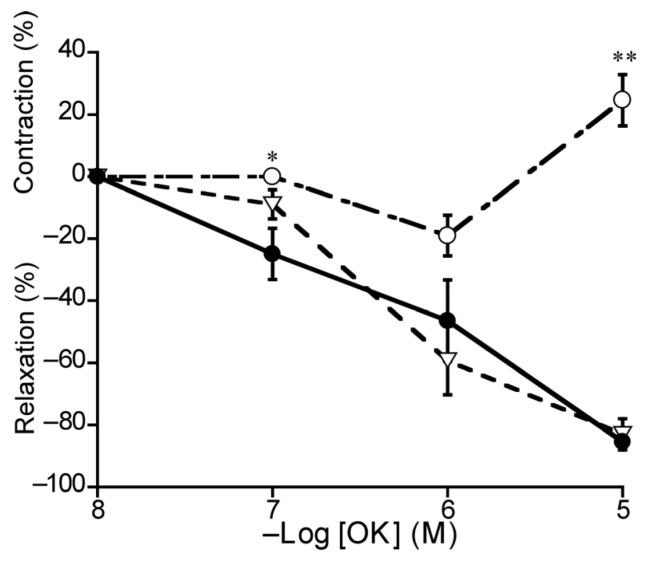
Effects of L-NNA (○: 10^−4^ M) and L-NNA plus indomethacin (10^−5^ M) (▽) on ornithokinin (OK)-induced relaxation (●) in duck basilar arteries. The contraction induced with 60 mM KCl and the relaxation induced with 10^−4^ M sodium nitroprusside were taken as 100% contraction and relaxation, respectively. Each point represents the mean ± SEM of 4 ducks. (* *p* < 0.05, ** *p* < 0.01 vs. control).

**Table 1 animals-13-02740-t001:** Effects of agonists on cerebral arteries in different animal species.

Agonists	Effect (Receptor Subtype)	Species
Noradrenaline	Contraction (α)	Chicken [12]
	Relaxation (β_1_, β_2_)	Pig [21]
Acetylcholine	Relaxation (M_3_)	Chicken [10], Cat [22]
5-Hydroxytryptamine	Contraction (5-HT_1_, 5-HT_2_)	Chicken [10], Rat [23]
Histamine	Contraction (H_1_)	Pig [24], Cattle [25]
	Relaxation (H_1_, H_2_)	Chicken [11]
Angiotensin II	Contraction (AT_1_)	Bat [26], Mouse [27]
Bradykinin	Contraction and relaxation (B_2_)	Pig [28]
	Relaxation (B_2_)	Mouse [29]

**Table 2 animals-13-02740-t002:** The pEC_50_ values and maximal response (E_max_) for agonists.

Agonists	pEC_50_	E_max_ (%)
**Resting tension**		
5-Hydroxytryptamine	5.84 ± 0.06	143.2 ± 9.3 ^a^
Histamine	4.37 ± 0.09	104.3 ± 14.0 ^a^
Noradrenaline	-	No response
Noradrenaline + Propranolol	-	No response
Angiotesin II	-	No response
**Precontracted condition**		
Acetylcholine	6.18 ± 0.17	−81.0 ± 5.4 ^b^
Noradrenaline	6.28 ± 0.05	−69.7 ± 3.7 ^b^
Noradrenaline + Phentolamine	6.30 ± 0.11	−68.8 ± 6.3 ^b^
Ornithokinin	6.20 ± 0.25	−85.4 ± 2.7 ^b^
Isoproterenol	7.22 ± 0.11	−82.8 ± 9.2 ^b^
Adrenaline	5.89 ± 0.10	−60.9 ± 9.5 ^b^

^a^ The contraction induced with 60 mM KCl was taken as 100%. ^b^ The relaxation induced with 10^−4^ M sodium nitroprusside was taken as 100%. Each value represents the mean ± SEM of 4–6 ducks. The concentration of propranolol and phentolamine is 10^−5^ M.

## Data Availability

The data presented in this study are available on request from the corresponding author.

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
