# Peer review of "Reduced Nitric Oxide Synthase Involvement in Aigamo Duck Basilar Arterial Relaxation"

_animals, 2023, doi:10.3390/ani13172740_

Round 1
Reviewer 1 Report
Nitric oxide (NO), which is produced by the endothelium, was discovered as the calming substance in the 1980s. Gaseous transmitters and their function in the control of the circulatory system have attracted attention as a result of this discovery. It has recently been clear that carbon monoxide (CO) and hydrogen sulphide (H2S) are also engaged in physiological and pathological processes in the cardiovascular system. Nebivolol, a third-generation b-blocker with vasodilating characteristics through increasing NO bioavailability, and nitrates, which have been used for more than a century, give solid evidence that chemicals acting via the NO route will continue to be a significant class of cardiovascular medications. However, veterinary data are still lacking, especially in the avian medicine
NO functions as a neuromodulator and collaborates with 5-hydroxytryptamine (5-HT), histamine (His), angiotensin II (Ang II), noradrenaline (NA), acetylcholine (ACh), and the avian bradykinin ornithokinin (OK). It also regulates blood vessel tone and subsequently blood pressure. Because all of these hormones are species-specific, it should be made clear that human hormones, not bird-specific hormones, were employed in this study.
Only Aigamo ducks (Anas platyrhynchos/Anas 48 platyrhynchos var. domesticus hybrid; limitation should state that no data on species-pure ducks are available should be included.
There are no quantitative data for the course to show how many percent of ducks reacted or how much BA was impacted. Data needs to be better.
L-arginine is a substrate in the production of guanidinoacetate in the kidneys and pancreas, which is subsequently transformed into creatine (the primary metabolite) in the liver; — into NO in endothelial cells of blood arteries.
Please let me know if blood tests have been done to rule out kidney liver disease? Whether the level of L-arginine was within the reference range. All of these components have an impact on hormone metabolism, which alters the experience's course.
Please provide more details about the possibility of a different chicken and duck course, and list the factors that would affect such a course.
NO citations in veterinary medicine can be improved.
Author Response
To Reviewer-1
Comments and Suggestions
Nitric oxide (NO), which is produced by the endothelium, was discovered as the calming substance in the 1980s. Gaseous transmitters and their function in the control of the circulatory system have attracted attention as a result of this discovery. It has recently been clear that carbon monoxide (CO) and hydrogen sulphide (H2S) are also engaged in physiological and pathological processes in the cardiovascular system. Nebivolol, a third-generation b-blocker with vasodilating characteristics through increasing NO bioavailability, and nitrates, which have been used for more than a century, give solid evidence that chemicals acting via the NO route will continue to be a significant class of cardiovascular medications. However, veterinary data are still lacking, especially in the avian medicine.
NO functions as a neuromodulator and collaborates with 5-hydroxytryptamine (5-HT), histamine (His), angiotensin II (Ang II), noradrenaline (NA), acetylcholine (ACh), and the avian bradykinin ornithokinin (OK). It also regulates blood vessel tone and subsequently blood pressure. Because all of these hormones are species-specific, it should be made clear that human hormones, not bird-specific hormones, were employed in this study.
Response:
5-hydroxytriptamine (serotonin), histamine, noradrenaline, and acetylcholine have been detected, and their functions described, in various non-mammalian vertebrates including avian species; however, bradykinin and angiotensin II may have bird-specific structure due to the relevant peptide. Accordingly, we used ornithokinin, which is regarded as avian bradykinin, in this study. In the notation, we have changed “angiotensin II” to “angiotensin II (human)” in the section of Reagents. The references listed below have been added, and are now cited in the text.
Fujita T, Aoki N, Mori C, Homma KJ, Yamaguchi S. Molecular biology of serotonergic systems in avian brains. Front Mol Neurosci. 2023 Jul 19;16:1226645. doi: 10.3389/fnmol.2023.1226645. PMID: 37538316; PMCID: PMC10394247.
Miki H, Inagaki N, Yamatodani A, Wada H. Regional distribution of histamine in the brain of non-mammalian vertebrates. Brain Res. 1992 Jan 31;571(1):129-32. doi: 10.1016/0006-8993(92)90518-e. PMID: 1611484.
Perry SF, Capaldo A. The autonomic nervous system and chromaffin tissue: neuroendocrine regulation of catecholamine secretion in non-mammalian vertebrates. Auton Neurosci. 2011 Nov 16;165(1):54-66. doi: 10.1016/j.autneu.2010.04.006. Epub 2010 Jun 12. PMID: 20547474.
Wächtler K. The regional production of acetylcholine in the brains of lower and higher vertebrates. Comp Biochem Physiol C Comp Pharmacol. 1980;65C(1):1-16. doi: 10.1016/0306-4492(80)90038-6. PMID: 6102008.
Comments and Suggestions
Only Aigamo ducks (Anas platyrhynchos/Anas 48 platyrhynchos var. domesticus hybrid; limitation should state that no data on species-pure ducks are available should be included.
Response:
We have added the following sentence in accordance with your suggestion. “Our data were obtained from Aigamo ducks, not from species-pure ducks, and this represents a limitation of this study.” (lines 94-96) Furthermore, we have added the term "Aigamo ducks" to the title.
Comments and Suggestions
There are no quantitative data for the course to show how many percent of ducks reacted or how much BA was impacted. Data needs to be better.
Response:
Since all results show similar responses to vasoactive substances, they are presented as the mean ± SEM and number of ducks used. To make it clearer, we noted in lines 169 and 170 that the percentage of reacting vessels is 100%.
Comments and Suggestions
L-arginine is a substrate in the production of guanidinoacetate in the kidneys and pancreas, which is subsequently transformed into creatine (the primary metabolite) in the liver; — into NO in endothelial cells of blood arteries.
Please let me know if blood tests have been done to rule out kidney liver disease? Whether the level of L-arginine was within the reference range. All of these components have an impact on hormone metabolism, which alters the experience's course.
Response:
The Reviewer’s suggestion would be crucial for research in clinical cases and/or experiment animals, but in this experiment, we only evaluated vasculature samples provided after the ducks had been slaughtered for processing as food animals; hence no blood tests for hepatic or renal function were, or could be, performed. Similar studies (with tissue samples but no blood tests) have been reported in other species.
Comments and Suggestions
Please provide more details about the possibility of a different chicken and duck course, and list the factors that would affect such a course.
Response:
The following sentence was added to lines 361-365.
These physiological differences could be explained by the earlier evolutionary emergence of ducks than chickens, which means that ducks may exhibit simpler endothelial physiology. In contrast, chickens have undergone significant adaptations to suit terrestrial life, leading to more complex functionalities mediated by the endothelium.
Comments and Suggestions
NO citations in veterinary medicine can be improved.
Response:
We agree that additional citations from the veterinary field would improve this manuscript, but so far we have been unable to identify any such appropriate references. If the reviewer is aware of any that we should consider in this manuscript, we would be delighted to include them and comment on them. Currently, about two thirds of the studies cited refer to research in one or more animal species, and we feel this may not be an unusual number for paper on pharmacology in a hitherto unevaluated species.
Reviewer 2 Report
The authors have examined effects of a series of agonists on contraction and relaxation of duck basilar arteries. The studies appear to be well designed and nicely executed. Unfortunately, the manuscript is difficult to follow and the title is misleading as the authors did not measure nitric oxide.
1. The authors need to have a native English speaker or editing service to go through the entire manuscript to improve the standard of writing.
2. Section headings throughout need to be clear and descriptive without acronyms.
3. Statements that provide no information such as “real time” (section 3.1.) appear to be gratuitous.
4. A table listing the agonists, their effects and the rationale for their use would be very useful.
5. Figure 1 Why is no y axis defined?
6. Figure 1 Legend Why is the statement “Relaxation induced by sodium nitroprusside was taken as 100%, in Figures 2, 5 and 8” in the legend.
7. Figure legends with two sections do not have overall titles covering both figures.
8. Figure 3. The authors have “o” signifying both 10-6 methiothepin and ketanserin.
See above
Author Response
To Reviewer-2
Comments and Suggestions for Authors
The authors have examined effects of a series of agonists on contraction and relaxation of duck basilar arteries. The studies appear to be well designed and nicely executed. Unfortunately, the manuscript is difficult to follow and the title is misleading as the authors did not measure nitric oxide.
Response:
Thank you for your comment. The nitric oxide synthase inhibitor L-NNA shifted the concentration-response curve for isoproterenol to the right in the chicken basilar artery but did not affect the curve in the duck basilar artery. Still, L-NNA-induced contraction (30.5%) accompanied by spontaneous nitric oxide release in ducks, which was markedly weaker than that (122.1%) in chickens. However, according to a reviewer comment, the title has been changed to “Reduced nitric oxide synthase involvement in Aigamo duck basilar arterial relaxation”.
- The authors need to have a native English speaker or editing service to go through the entire manuscript to improve the standard of writing.
Response:
One of our coauthors, a native British-English speaker, has checked the manuscript to improve the standard of writing. If you have specific concerns that need addressing, please let us know and we can review this manuscript with the relevant focus.
- Section headings throughout need to be clear and descriptive without acronyms.
Response:
We have changed the headings of each section to avoid abbreviations and to provide a clearer explanation. The changed section headings are stated below.
- 1. Spontaneous nitric oxide and prostaglandin release
- 3. Involvement of 5-hydroxytriptamine receptor subtype
- 4. Involvement of histamine receptor subtype
- 5. Responsiveness to β-adrenergic receptor agonists
- 7. Involvement of muscarinic receptor subtype
- 8. Effects of nitric oxide synthase and cyclooxygenase inhibitors on ornithokinin-induced response
- Statements that provide no information such as “real time” (section 3.1.) appear to be gratuitous.
Response:
We have simplified the paragraph as follows:
Typical responses to L-NNA (a NOS inhibitor, 10-4 M) followed by indomethacin (a cyclooxygenase inhibitor, 10-5 M) under the resting tone are illustrated in Figure 1.
- A table listing the agonists, their effects and the rationale for their use would be very useful.
Response:
The agonists used in this study were intrinsic vasoactive substances commonly used in similar research. Table 1 lists the agonists used and their reported effects on basilar arteries in various species, which act as a rationale for their use in the current study.
Table 1. Effects of agonists on basilar arteries in different animal species.
|
Agonists |
Effect (receptor subtype) |
Species |
|
Noradrenaline
|
Contraction (α) Relaxation (β1, β2) |
Chicken [43] Pig [21] |
|
Acetylcholine |
Relaxation (M3) |
Chicken [17] |
|
5-Hydroxytryptamine |
contraction (5-HT1, 5-HT2) |
Chicken [17] |
|
Histamine
|
contraction (H1) relaxation (H1, H2) |
Pig [24], Cattle [25] Chicken [28] |
|
Angiotensin II |
contraction (AT1) |
Bat [10], Pig [26] |
|
Bradykinin
|
contraction and relaxation (B2) relaxation (B2) contraction (unconfirmed) |
Pig [20] Mouse [13] Horse [39] |
- Figure 1 Why is no y axis defined?
Response:
The unit of force (mN) was added to the y-axis.
- Figure 1 Legend Why is the statement “Relaxation induced by sodium nitroprusside was taken as 100%, in Figures 2, 5 and 8” in the legend.
Response:
We have removed this statement because it may have confused the reader.
- Figure legends with two sections do not have overall titles covering both figures.
Response:
Figure legends with two sections have been changed as described below.
Figure 2 legends: Responsiveness to 5-hydroxytriptamine (5-HT), histamine (His), noradrenaline (NA) and angiotensin (Ang) II under resting tension [A], and to NA, acetylcholine (ACh), and ornithokinin (OK) under a precontracted condition induced by 5-HT [B].
Figure 3 legends: Effects of 5-hydroxytriptamine (5-HT) antagonists on 5-HT-induced contraction in the duck basilar artery. The antagonists were methiothepin (Δ: 10-8 M, â–½: 10-7 M, â—‹: 10-6 M) [A], and ketanserin (x: 10-6 M) [B]. The Schild plot for methiothepin is shown in [C].
Figure 4. Effects of histamine (His) antagonists on His-induced contraction in the duck basilar artery. The antagonists were diphenhydramine (Δ: 10-7 M, â–½: 10-6 M, â–¡: 10-5 M) and cimetidine (â—‹: 10-5 M) [A]. The Schild plot for diphenhydramine is shown in [B].
Figure 6. Effects of β-adrenergic receptor antagonists and a NOS inhibitor (L-NNA) on isoproterenol-induced relaxation in the duck basilar artery. The antagonists or inhibitor were atenolol (â–²: 10-6 M), butoxamine (â–¼: 10-6 M), L-NNA (â—‹: 10-4 M) [A], and SR 59230A (Δ: 10-7 M, ×: 10-6 M) [B]. The Schild plot for SR 59230A is shown in [C].
Figure 7. Effects of acetylcholine (ACh) antagonists and a NOS inhibitor (L-NNA) on ACh-induced relaxation in the duck basilar artery. The antagonists or inhibitor were atropine (â—‹: 10-8 M, Δ: 10-7 M) [A], pirenzepine (â– : 10-6.5 M, â–¡: 10-6 M) [B], hexahydro-sila-difenidol hydrochloride, p-fluoroanalog (pFHHSiD; â–²: 10-7 M, ×: 10-6 M) [C], and L-NNA (â—‡:10-4 M) [D]. The Schild plots for pirenzepine and pFHHSiD were shown in [E] and [F], respectively.
- Figure 3. The authors have “o” signifying both 10-6methiothepin and ketanserin.
Response:
In figure 3, the mark of “o” signifying ketanserin (10-6 M) has been changed to ”x”.